# Dendrimer-Conjugated nSMase2 Inhibitor Reduces Tau Propagation in Mice

**DOI:** 10.3390/pharmaceutics14102066

**Published:** 2022-09-28

**Authors:** Carolyn Tallon, Benjamin J. Bell, Anjali Sharma, Arindom Pal, Medhinee M. Malvankar, Ajit G. Thomas, Seung-Wan Yoo, Kristen R. Hollinger, Kaleem Coleman, Elizabeth L. Wilkinson, Sujatha Kannan, Norman J. Haughey, Rangaramanujam M. Kannan, Rana Rais, Barbara S. Slusher

**Affiliations:** 1Johns Hopkins Drug Discovery, Baltimore, MD 21205, USA; 2Department of Neurology, Johns Hopkins University School of Medicine, Baltimore, MD 21205, USA; 3Center for Nanomedicine, Department of Ophthalmology, Wilmer Eye Institute, Johns Hopkins University School of Medicine, Baltimore, MD 21231, USA; 4Department of Cell Biology, Johns Hopkins School of Medicine, Baltimore, MD 21205, USA; 5Moser Center for Leukodystrophies at Kennedy Krieger, Kennedy Krieger Institute, Anesthesiology and Critical Care Medicine, Johns Hopkins University School of Medicine, Baltimore, MD 21205, USA; 6Hugo W. Moser Research Institute at Kennedy-Krieger Inc., Baltimore, MD 21205, USA; 7Psychiatry and Behavioral Science, Johns Hopkins University School of Medicine, Baltimore, MD 21205, USA; 8Center for Nanomedicine at the Wilmer Eye Institute, Department of Chemical and Biomolecular Engineering, Johns Hopkins University School of Medicine, Baltimore, MD 21205, USA; 9Department of Pharmacology and Molecular Sciences, Johns Hopkins School of Medicine, Baltimore, MD 21205, USA; 10Johns Hopkins Drug Discovery, Johns Hopkins School of Medicine, Baltimore, MD 21205, USA; 11Department of Oncology, Johns Hopkins School of Medicine, Baltimore, MD 21205, USA

**Keywords:** Alzheimer’s disease, extracellular vesicles, tau, dendrimer, neutral sphingomyelinase 2, ceramide, DPTIP, PDDC

## Abstract

Alzheimer’s disease (AD) is characterized by the progressive accumulation of amyloid-β and hyperphosphorylated tau (pTau), which can spread throughout the brain via extracellular vesicles (EVs). Membrane ceramide enrichment regulated by the enzyme neutral sphingomyelinase 2 (nSMase2) is a critical component of at least one EV biogenesis pathway. Our group recently identified 2,6-Dimethoxy-4-(5-Phenyl-4-Thiophen-2-yl-1H-Imidazol-2-yl)-Phenol (DPTIP), the most potent (30 nM) and selective inhibitor of nSMase2 reported to date. However, DPTIP exhibits poor oral pharmacokinetics (PK), modest brain penetration, and rapid clearance, limiting its clinical translation. To enhance its PK properties, we conjugated DPTIP to a hydroxyl-PAMAM dendrimer delivery system, creating dendrimer-DPTIP (D-DPTIP). In an acute brain injury model, orally administered D-DPTIP significantly reduced the intra-striatal IL-1β-induced increase in plasma EVs up to 72 h post-dose, while oral DPTIP had a limited effect. In a mouse tau propagation model, where a mutant hTau (P301L/S320F) containing adeno-associated virus was unilaterally seeded into the hippocampus, oral D-DPTIP (dosed 3× weekly) significantly inhibited brain nSMase2 activity and blocked the spread of pTau to the contralateral hippocampus. These data demonstrate that dendrimer conjugation of DPTIP improves its PK properties, resulting in significant inhibition of EV propagation of pTau in mice. Dendrimer-based delivery of DPTIP has the potential to be an exciting new therapeutic for AD.

## 1. Introduction

Alzheimer’s disease (AD) is a progressive neurodegenerative disease characterized by the accumulation of amyloid-β (Aβ) plaques and hyperphosphorylated tau (pTau)-containing neurofibrillary tangles within the brain. Memory impairment is often the first sign of AD, yet the underlying disease pathology develops many years before cognitive and other behavioral phenotypes manifest [1,2]. While several approved medications were designed to treat cognitive symptoms associated with AD [3] and a recently approved treatment targets AD-associated amyloid pathology [4], there are no approved therapeutics which target the accumulation of pTau. This is important because while Aβ deposition does not correlate strongly to the incidence of cognitive phenotypes [5], tau imaging studies have demonstrated a relationship between brain tau levels and the emergence and progression of cognitive impairments in AD [6,7]. This highlights the need for developing novel therapeutics that slow pTau accumulation and spread.

Extracellular vesicles (EVs) are small, heterogeneous membranous structures that facilitate intercellular communication via the delivery of a repertoire of cargo, including nucleic acids, lipids, and proteins [8]. While multiple pathways exist for EV biogenesis [9], one pathway is facilitated by the enrichment of membrane ceramides which have small polar head groups and negative curvature [10]. As such, several groups have shown that EV release can be regulated, in part, via the enzyme neutral sphingomyelinase 2 (nSMase2), which catalyzes the formation of ceramide from sphingomyelin [8,10,11]. Inhibition of nSMase2 has been identified as a therapeutic target for several diseases, including neurological disorders such as AD, amyotrophic lateral sclerosis, multiple sclerosis, and Parkinson’s disease (for review, see [8]). Elevated brain ceramide levels and elevated cerebral spinal fluid (CSF) EV levels have been observed in samples from AD patients vs. controls [12,13,14], and a growing body of evidence supports the hypothesis that pathological pTau spreads between neurons along anatomically connected pathways via EVs [15,16,17,18,19,20,21]. In fact, pTau spread has been shown to be reduced by nSMase2 knockdown and pharmacological inhibition using synaptosomes isolated from the brains of individuals with AD, as well as in tau-mediated mouse models of AD [16,22]. Due to the possible therapeutic benefit of inhibiting nSMase2 activity, there has been growing interest in discovering small molecule inhibitors that can be clinically translated. Several inhibitors have been identified to date; however, they suffer from low potency, poor oral bioavailability, limited brain penetration, metabolic instability, and insolubility, which has hampered their translation [8,23,24,25,26].

With this in mind, our laboratory carried out a human nSMase2 high-throughput screening (HTS) campaign which identified two clusters of confirmed hits. Through extensive structure–activity relationship studies and the synthesis of hundreds of analogs, the first series of hits was developed into the lead inhibitor phenyl(R)-(1-(3-(3,4-dimethoxyphenyl)-2,6-dimethylimidazo [1,2-b]pyridazin-8-yl)pyrrolidin-3-yl)-carbamate (PDDC) [27,28]. PDDC exhibits excellent oral bioavailability and brain penetration, however, its potency is modest (300 nM) and its dosing limited due to low solubility. The second HTS confirmed hit was 2,6-Dimethoxy-4-(5-Phenyl-4-Thiophen-2-yl-1H-Imidazol-2-yl)-Phenol (DPTIP), which was found to be potent (30 nM) and selective [28,29]. However, its translational potential is limited by poor oral bioavailability, modest brain penetration, and rapid clearance [30,31]. Our efforts to optimize DPTIP did not produce substantial improvements [31].

We have recently also employed drug delivery strategies to address these pharmacokinetic limitations. In the first strategy, we synthesized a series of prodrugs by masking DPTIP’s phenolic hydroxyl group, and demonstrated an improvement in plasma and brain exposures, as well as an increased plasma t_1/2_ [30]. While the prodrugs were improved vs. DPTIP, their oral bioavailability was still limited and their plasma t_1/2_ < 3 h, which is not ideal for a chronic therapy. Another strategy that has been used to deliver small molecules to the brain is conjugation to dendrimer nanoparticles. Dendrimers are emerging as promising candidates for imaging and targeted drug/gene delivery for several neuroinflammatory diseases [32,33,34,35]. Hydroxyl-terminated poly(amidoamine) (PAMAM) dendrimers are widely studied [36,37], shown to be nontoxic, even at doses > 500 mg/kg, and are cleared intact through the kidney [33,38,39,40]. Hydroxyl-PAMAM dendrimers have been demonstrated to be orally bioavailable, facilitate CNS penetration, and prolong the half-life of conjugated small molecules [41,42,43,44,45,46,47,48,49,50]. Previous research by our lab has shown that these dendrimers selectively localize in activated glia in the brain in small and large animal models, and can deliver drugs to the site of injury, thus producing positive therapeutic outcomes [33,48,50,51,52]. In fact, hydroxyl-dendrimer nanotherapy was recently shown to attenuate inflammation and neurological injury markers and improve outcomes in a phase 2a clinical trial in patients with severe COVID-19 [53]. We therefore aimed to improve the oral bioavailability, half-life, and brain delivery of DPTIP by conjugating it to a hydroxyl-PAMAM dendrimer delivery system (D-DPTIP).

Herein, D-DPTIP was successfully synthesized and characterized. A single oral dose of D-DPTIP inhibited EV release from the brain into plasma up to 72 h post administration in an acute brain injury model. In addition, chronic D-DPTIP administration inhibited brain nSMase2 enzymatic activity and reduced the spread of hyperphosphorylated tau in an AAV-mediated tau propagation murine model.

## 2. Materials and Methods

### 2.1. Synthesis of D-DPTIP Conjugates

All reactions were carried out in standard flame-dried glassware unless otherwise stated. The solvents for reactions and purification, as well as the deuterated solvents for NMR spectroscopy, were purchased from Sigma Aldrich (St. Louis, MO, USA). DPTIP (compound **1**) was synthesized as we previously reported [29]. 1-[3-(Dimethylamino)propyl]-3-ethylcarbodiimide methiodide (EDC), 4(dimethylamino)pyridine (DMAP), copper sulfate pentahydrate, sodium ascorbate, and 5-hexynoic acid were purchased from Sigma Aldrich (St. Louis, MO, USA). Azido-PEG-4-acid was purchased from Broad Pharm (San Diego, CA, USA). Pharmaceutical grade ethylenediamine-core generation 4 polyamidoamine (PAMAM-OH) dendrimer (compound **3**) was received from Dendritech (Midland, MI, USA) in the form of a methanolic solution. The methanol was evaporated before use to afford the dendrimer as a white hygroscopic solid. The purification of the dendrimer conjugates was carried out using 1 kDa dialysis membrane purchased from Spectrum Laboratories Inc. (New Brunswick, NJ, USA). Synthesis and characterization (HPLC and NMR spectra) of the intermediate compounds **2** and **4** are described in the Appendix A using previously described methods (Appendix A) [44,54,55]. Compound **5** (D-DPTIP) was synthesized by stirring a solution of compound **4** (2 g, 0.131 mmoles) and compound **2** (685 mg, 1.05 mmoles) in DMF, then CuSO_4_.5H_2_O (8.4 mg, 0.03 mmoles) dissolved in 1 mL water was added, followed by the addition of sodium ascorbate (13.4 mg, 0.06 mmoles) dissolved in 1 mL water. The reaction was carried out at 40 °C in a microwave reactor for 6 h. Upon completion, the reaction mixture was diluted with DMF and dialyzed against DMF. This was followed by dialysis in EDTA solution, followed by water dialysis. The aqueous solution was then lyophilized to afford the pure product as a white solid. Yield: 89%. 1H NMR (500 MHz, D_2_O) δ 8.12–7.73 (D-amide H, triazole H), 7.59–6.92 (m, DPTIP ArH), 4.71 (bs, D–OH), 4.46 (t, linker–CH_2_), 4.01 (t, linker–CH_2_), 3.84 (DPTIP–OCH_3_), 3.80–3.70 (m, linker–CH_2_), 3.63–3.45 (m, D–CH_2_), 3.41–3.22 (m, D–CH_2_, DPTIP H, PEG linker H), 3.20–3.29 (m, D–CH_2_), 2.86–2.77 (m, –CH_2_), 2.76–2.53 (m, D–CH_2_), 2.49–2.07 (m, D–CH_2_), 1.89–1.78 (m, linker–CH_2_) (Appendix A).

HPLC: Retention time: 12.05 min, purity: 99.1% (Appendix A).

### 2.2. High Performance Liquid Chromatography (HPLC)

The purity and the drug release profile of D-DPTIP were evaluated using HPLC. The HPLC system (Waters Corporation, Milford, MA, USA) was equipped with a 2998 photodiode array detector, a 1525 binary pump, an in-line degasser AF, and Waters Empower software Empower TM 3 (Build 3471) (Waters Corporation, Milford, MA, USA). The purity was analyzed using a C18 symmetry column at 210 or 280 nm using a gradient flow. A mobile phase consisting of 0.1% TFA and 5% ACN in water (buffer A) and 0.1% TFA in ACN (buffer B) was used. The gradient began from 100:0 (A:B) and increased gradually to 10:90 (A:B) at 20 min, maintained at 10:90 (A:B) at 25, moved back to 90:10 (A:B) at 26 min, and finally returned to 100:0 (A:B) at 30 min. A flow rate of 1 mL/min was maintained throughout the run.

### 2.3. Mass Spectroscopy

Accurate mass measurements (HRMS) were performed on a Bruker microTOF-II mass spectrometer using ESI in positive mode and direct flow sample introduction in a CH_3_CN:H_2_O (9:1) solvent system. Either protonated molecular ions [M + nH]n+ or adducts [M + nX]n+ (X = Na, K, NH4) were used for empirical formula confirmation.

### 2.4. Nuclear Magnetic Resonance (NMR) Spectroscopy

The structures of intermediates and dendrimer conjugates were characterized using 1H NMR spectroscopy. The 1H NMR spectra were recorded on a Bruker spectrometer at ambient temperatures at 500 MHz. The chemical shifts were reported in ppm. The chemical shifts of residual protic solvents [CDCl_3_ (1H, δ = 7.27 ppm; 13C, δ = 77.0 ppm (central resonance of the triplet)), D_2_O (1H, δ = 4.79 ppm); and DMSO-d6 (1H, δ = 2.50 ppm)] were used for chemical shifts calibration. The multiplicities of the proton peaks are abbreviated as brs = broad singlet, d = doublet, t = triplet, q = quartet, and m = multiplet.

### 2.5. Dynamic Light Scattering (DLS) and Zeta Potential (ζ)

The size and the zeta potential distribution of the D-DPTIP were recorded using Malvern Zetasizer (Malvern Instrument Ltd., Worchester, UK). The instrument was equipped with a 50 mW He-Ne laser (633 nm). For size measurements, the D-DPTIP solution was prepared in deionized water at a concentration of 0.1 mg/mL. The solution was filtered through 0.2 µm syringe filters (Pall Corporation, 0.2 µm HT Tuffryn membrane) before performing the size measurements. A UV transparent disposable cuvette with dimensions of 12.5 × 12.5 × 45 mm (SARSTEDT) was used for size measurement. The measurements were recorded in triplicates. For zeta potential measurements, the D-DPTIP solution was prepared in 10 mM NaCl at a concentration of 0.1 mg/mL, and the solution was filtered through 0.2 µm syringe filters (Pall Corporation, 0.2 µm HT Tuffryn membrane). The zeta potential measurements were also performed in triplicates. A Malvern Zetasizer Nanoseries disposable folded capillary cell (Malvern Panalytical, Westborough, MA, USA) was used.

### 2.6. In Vitro Drug Release

The release of free DPTIP from D-DPTIP was evaluated in PBS (pH 7.4) in an attempt to mimic plasma conditions, and using citrate buffer with esterase (pH 5.5) to mimic intracellular/endosomal conditions following our previously published protocol [54,55]. The conjugate was dissolved in each buffer at a concentration of 1 mg/mL. The solutions were prepared in duplicates for each condition and incubated at 37 °C with constant shaking. At specified time points, 200 µL aliquots were taken from each buffer solution in duplicates and quenched with 100 µL of methanol. The samples were stored at −20 °C until further analysis using HPLC. A standard curve was obtained on HPLC for free DPTIP concentrations, and the release of free DPTIP from the conjugate was calculated for the samples at specified time-points from the calibration curve.

### 2.7. Animal Studies

All experiments and animal care carried out in these studies were in accordance with the Johns Hopkins University Animal Care and Use Committee guidelines. All animal studies carried out in this manuscript were reviewed for ethical use of animals and approved by the Johns Hopkins University Animal Care and Use Committee (protocol #MO20M71; approved on 18 March 2020). The 2–3-month-old GFAP-GFP transgenic mice (for in vivo IL-1β EV release model) and 8-week old C57BL/6J mice (for AAV-hTau injection model) were purchased from Jackson Laboratories (strain #010835 and #000664, respectively). For the EV release assay, male mice were used to avoid the effects of female specific hormones (i.e., estrogen) on inflammation [56]. For the studies using the AAV-hTau model, equal male and female mice were used and assessed separately. Animals were group housed in a 15 h light/9 h dark cycle, and the temperature was controlled at 21 °C with 42% humidity and with ad libitum access to food and water. Body weights were measured prior to AAV-hTau injection and served as the baseline measurement. Weekly body weights were recorded for each mouse throughout the entire treatment period.

### 2.8. DPTIP and D-DPTIP Preparations for In Vivo Studies

DPTIP was administered orally at a dose of 100 mg/kg in a 5% DMSO, 10% Tween-80 in saline solution. D-DPTIP was administered orally at a dose of 100 mg/kg DPTIP equivalent dissolved in drinking water. Vehicle control consisted of administering an empty dendrimer at an equivalent concentration to the D-DPTIP dose also dissolved in drinking water. DPTIP, D-DPTIP, and empty vehicle doses were given either as a single bolus (for EV release assays) or three times per week (for efficacy studies) via oral gavage.

### 2.9. Brain Levels of DPTIP Following Oral D-DPTIP in AD Mice

Studies were conducted to select a dose of D-DPTIP for mouse efficacy studies that achieved DPTIP levels in the brain greater than its nSMase2 IC_50_. Briefly, D-DPTIP was administered as a single oral gavage in PS19 mice (n = 3) at a dose of 30 mg/kg and 100 mg/kg DPTIP equivalent, dissolved in drinking water. The animals were euthanized (using CO_2_) 24 h post D-DPTIP administration and the brains were dissected, immediately flash frozen (−80 °C), and stored at −80 °C until LC/MS-MS analysis.

### 2.10. DPTIP Bioanalysis

Quantitation of DPTIP released from D-DPTIP was performed using our published DPTIP bioanalytical LC/MS/MS method [30], with minor modifications. Briefly, calibration standards of DPTIP were prepared using naïve brain tissue from untreated mice. For quantifying DPTIP, brain tissues were diluted 1:5 *w/v* with acetonitrile containing losartan (0.5 μM) and homogenized, followed by vortex-mixing and centrifugation at 10,000× *g* for 10 min at 4 °C. A 50 μL aliquot of the supernatant was diluted with 50 μL of water and transferred to 250 μL polypropylene autosampler vials sealed with teflon caps. Then, 2 μL of the sample were injected into the LC/MS/MS system for analysis. Chromatographic analysis was performed using an Accela ultra-high-performance system consisting of an analytical pump and an autosampler coupled with a TSQ Vantage mass spectrometer. Separation was achieved using Agilent Eclipse Plus column (100 × 2.1 mm i.d.) packed with a 1.8 μm C18 stationary phase. The mobile phase consisted of 0.1% formic acid in acetonitrile and 0.1% formic acid in water. The [M + H]^+^ ion transition of DPTIP (*m/z* 378.956 → 363.073, 200.055) and losartan (IS) (*m/z* 423.200 → 207.107, 180.880) were used. Concentrations (nmol/g) of released DPTIP in the brain samples were determined and plots of mean concentration vs. time were constructed.

### 2.11. IL-1β Induced EV Secretion in an Acute Brain Injury Model

IL-1β striatal injections and EV measurements were performed as previously described by our group [27,57]. Briefly, mice were deeply anesthetized with 1–2% isoflurane (Primal Critical Care, Inc., Bethlehem, PA, USA) and a small burr hole was made in the skull over the left striatum. A total volume of 3 µL of IL-1β (0.1 ng/3 μL) was injected via a pulled glass capillary (tip diameter < 50 μm) at a rate of 0.5 µL·min^−1^ into the left striatum at the stereotaxic co-ordinates A/P + 1; M/L − 2; and −3 D/V using bregma as a reference. Control mice were given sterile saline injections. The capillary was kept in place for 5 min following the infusion to allow the solution to diffuse into the tissue. Mice were separated into 6 treatment groups (n = 4/group): (1) saline injection + 48 h oral vehicle pretreatment; (2) IL-1β injection + 48 h oral vehicle pretreatment; (3) IL-1β injection + 48 h oral 100 mg/kg DPTIP pretreatment; (4) IL-1β injection + 48 h oral 100 mg/kg DPTIP equivalent dose of D-DPTIP (D-DPTIP e.q.) pretreatment; (5) IL-1β injection + 72 h oral 100 mg/kg D-DPTIP e.q. pretreatment; (6) IL-1β injection + 96 h oral 100 mg/kg D-DPTIP e.q. pretreatment. Mice were sacrificed via an overdose of isoflurane (Primal Critical Care, Inc., Bethlehem, PA, USA) 4 h after the IL-1β injection and their blood was collected via a cardiac puncture using heparin-coated needles (Sigma) into EDTA-coated tubes (BD). Blood was immediately centrifuged at 2700× *g* for 15 min at 20 °C to obtain plasma which was subsequently spun at 10,000× *g* for 15 min at 4 °C to obtain platelet-free plasma and remove large particles such as apoptotic bodies. To isolate GFP + EVs, plasma was incubated at 4 °C overnight with 2 × 10^7^ Dynabeads M-450 Epoxy (Invitrogen, Waltham, MA, USA) coupled with an anti-GFP antibody (Thermo Fisher, Waltham, MA, USA) at a ratio of 200-μg antibody per 4 × 10^8^ beads. The beads were washed and EVs bound to the anti-GFP Dynabeads were separated by being placed on a magnet prior to being eluted using 0.1 M glycine (pH 3.0).

### 2.12. Quantitation of Plasma EVs in IL-1β Induced Acute Brain Injury Model

Plasma EVs and isolated GFP+ EVs from the IL-1β EV release mouse model were quantified using the ZetaView Nanoparticle Tracker (Particle Metrix GmBH, Meerbusch, Germany) corresponding ZetaView software (8.05.14.SP7) as previously described [27,58]. Briefly, 1 mL of 10,000× *g* centrifuged plasma was injected into the sample-carrier cell and the particle count was measured at five different positions with two cycles of reading at each position. Pre-acquisition parameters were set at 23 °C, with a sensitivity of 65, and a frame rate of 30 frames/s with a shutter speed and laser pulse duration of 100 ms. Post-acquisition parameters were set to a minimum brightness of 25 with a maximum pixel size of 200 pixels and a minimum pixel size of 10 pixels. The sample-carrier cell was washed with 1X PBS after each sampling. EV concentrations were determined from three independent experiments performed on each sample.

### 2.13. Stereotaxic Injection of AAV-hTau Vector

The hTau vector CBA-hTau24(P301L)(S320F)-WPRE was kindly provided by the Chakrabarty lab (University of Florida, Gainesville, FL, USA) [59] and packaged into AAV1-serotype viral particles by Vector Biolabs (Philadelphia, PA, USA). These AAV vectors containing the human P301L/S320F double mutant tau plasmids (AAV-hTau) were stereotaxically injected into the left hippocampus of 10-week old mice using a modified version of previously reported protocols [16,59]. Specifically, mice were deeply anesthetized using 1–2% isoflurane and placed on a heating pad to keep their body temperature at 37 °C, and an optical gel was placed over their eyes to prevent drying out. The mice were secured in a stereotaxic apparatus (Stoelting, Wood Dale, IL, USA); a small incision was made in the skin along the midline from approximately lambda to bregma and the skull was exposed and cleaned with hydrogen peroxide. A small burr-hole was made in the skull using a Dremel at the coordinates AP-2.3, ML-2.1, and a pulled glass capillary needle (tip diameter < 50 μm) was lowered to DV-2.2 to inject into the CA3 region of the left dorsal hippocampus. A digital nanoinjector (Stoelting) attached to a mineral oil-filled 5 uL gas-tight syringe (Hamilton) drove the stereotaxic injection of 5 × 10^9^ viral particles in <250 nL PBS over the course of 5 min, and then the syringe was left in place for 5 min to allow for viral diffusion and to prevent backflow. After removal of the syringe, the incision was closed with cyanoacrylate glue (Vetbond, 3M) and the mouse provided ketoprofen analgesia. Mice were placed back in their home cage after recovery from anesthesia, and monitored for signs of distress or infection over the following 48 h. Mice were randomized into treatment and placebo groups after recovery.

### 2.14. Immunofluorescence Imaging in AAV-hTau Model

Six weeks following the initial injection, mice were sacrificed using an overdose of isoflurane (Primal Critical Care, Inc., Bethlehem, PA, USA) and animals were cardiac perfused with ice cold 1X PBS, followed by 2% paraformaldehyde (PFA; Electron Microscopy Sciences). Whole brains were carefully dissected and post-fixed in 2% PFA for 24 h at 4 °C. 2% PFA was chosen to reduce autofluorescence in the green channel. Brains were then cryopreserved in 30% sucrose prior to being snap frozen in Tissue-Tek O.C.T. compound (Sakura FineTek USA, Inc., Torrence, CA, USA) and stored at −80 °C. Brains were cryosectioned at a 20 µm thickness prior to being permeabilized and blocked with a 5% normal goat serum (Vector Laboratories, Burlingame, CA, USA), 0.3% Triton X-100 (Sigma Aldrich), and 1X PBS blocking solution for 1 h at room temperature. Sections were incubated with a primary antibody against Thr181 phosphorylated tau (Cell Signaling, Danvers, MA, USA) overnight at 4 °C. Sections were then incubated in an anti-rabbit AlexFluor 488 secondary antibody at room temperature for 1 h before being washed well in 1X PBS 5 times for 3 min. Sections were then stained with an AlexaFluor 647 conjugated anti-NeuN primary antibody (Abcam, Cambridge, UK) for 1 h at room temperature before being treated with Hoescht 33342 (Thermo Fisher) to identify nuclei. Sections were coverslipped with Prolong Glass antifade mountant (Invitrogen) prior to being imaged on an LSM 800 confocal microscope (Zeiss, Jena, Germany). The entire brain section was imaged using tiling and Z-stack modules with a 10X objective. Close up tiled images of the ipsilateral and contralateral dentate gyrus were acquired with a 20X objective. Images were then collapsed to visualize the entire brain section, and gamma and brightness were adjusted for optimal presentation.

### 2.15. Dendrimer-Cy5 Localization Studies

Six weeks following AAV-hTau injection, AAV-hTau mice were injected with a single oral 55 mg/kg dose of dendrimer-Cy5 conjugate, as previously described [44]. Twenty-four hours after the injection, mice were euthanized by an overdose of isoflurane and were cardiac perfused with 1X ice cold PBS, followed by ice cold 2% paraformaldehyde (PFA; Electron Microscopy Sciences). Brains were then dissected and post fixed for 24 h at 4 °C in 2% PFA before being cryoprotected in 30% sucrose and snap frozen in O.C.T. prior to being stored at −80 °C. Brains were then cryosectioned on a cryostat (COMPANY) at a thickness of 30 µm. Sections were permeabilized and blocked with a 5% normal goat serum, 0.3% triton X-100, 1X PBS blocking solution, and then stained with a primary antibody for Thr181 phosphorylated tau (Cell Signaling Technology) overnight at 4 °C. The sections were then incubated with the appropriate secondary antibody for 1 h at room temperature prior to being washed thoroughly 5 times with 1X PBS. The sections were then incubated with a SPICA Dye 568 conjugated primary antibody against Iba1 (Wako) overnight at 4 °C. The sections were then treated with Hoescht 33342 (Thermo Fisher) to stain nuclei prior to cover slipping with Prolong Glass antifade mountant. Slides were then imaged using an LSM 800 confocal microscope (Zeiss, Jena, Germany) either with a 20X objective and tiling and Z-stack modules for the broad overview, or a 40X objective for the close-up images. Z-stacks were collapsed, and images were processed for optimal presentation.

### 2.16. AAV-hTau Mice Dosing Regimen

Mice for nSMase2 activity and efficacy studies were injected with the AAV-hTau vector at 10 weeks of age and randomly assigned either to a vehicle or 100 mg/kg D-DPTIP equivalent dose group. Five mice per sex were assigned to each group for nSMase2 activity. Efficacy groups had 10 mice per sex assigned to either the vehicle or D-DPTIP groups. All mice were orally dosed 3 times per week and their body weights were measured weekly. For efficacy assessments, 4 vehicle male, 6 D-DPTIP male, 5 vehicle female, and 4 D-DPTIP female mice were removed from assessment due to incorrect injection location.

### 2.17. nSMase2 Activity Assessments in AAV-hTau Mice

After 6 weeks of dosing, mice were sacrificed using an overdose of isoflurane and cardiac perfused with ice cold 1X PBS. Brains were quickly dissected, and the left hippocampus and frontal cortex was micro-dissected. All tissues were snap frozen in liquid nitrogen. The fluorescence-based assay for measuring ex vivo nSMase2 activity was employed as described previously [24].

### 2.18. Efficacy Analysis in AAV-hTau Mice

After 6 weeks of dosing, mice were sacrificed using an overdose of isoflurane and cardiac perfused with ice cold 1X PBS followed by 2% PFA. Brains were dissected and post-fixed in 2% PFA for 24 h at 4 °C, prior to being cryopreserved in 30% sucrose and snap frozen in O.C.T. and stored at −80 °C. Brains were then cryosectioned on a cryostat at a thickness of 20 µm. Sections were stained with a primary antibody against Thr181 phosphorylated tau at 4 °C overnight. Sections were then incubated with the appropriate secondary for 1 h at room temperature prior to being washed well 5 times for 3 min. Sections were then incubated with an AlexaFluor 647 conjugated anti-NeuN primary antibody for 1 h at room temperature before being treated with Hoescht 33342 and cover slipped with Prolong Glass antifade mountant. The ipsilateral and contralateral dentate gyrus in the dorsal hippocampus from 5 sections spaced 120 µm apart were imaged using an LSM 800 confocal microscope (Zeiss). All images were acquired using identical imaging parameters (camera gain, laser power, pinhole size, pixel count, etc.) and only raw images were analyzed for mean fluorescence intensity. Each section was treated as a replicate. Tau fluorescence measurements: The mean fluorescence intensity (MFI) of both the ipsilateral and contralateral dentate gyrus per section were measured using the Zen Blue 2.3 Software (Carl Zeiss Microscopy Gmbh, Munich, Germany). The ratio of the contralateral to ipsilateral MFI was calculated to account for variability in the concentration of the AAV-hTau particles or injection volume between animals. Animals where the ipsilateral injection was too lateral or too ventral such that the dentate gyrus of the dorsal hippocampus did not take up robust levels of the AAV-hTau were excluded from analysis. Tau positive neurons quantification: The total number of NeuN positive cells in the hilus region of the contralateral dentate gyrus from the same images used for the MFI assessments were counted. The number of NeuN/Thr181 Tau double positive cells were also quantified. The percentage of all NeuN positive cells that were double positive were then calculated.

### 2.19. Clinical Chemistries

After the 6-weeks of treatment for the efficacy studies, blood was collected via cardiac puncture from the mice prior to perfusion and placed in an uncoated blood collection tube. Whole blood was allowed to coagulate at room temperature for 30 min prior to being centrifuged at 1500× *g* for 10 min at 4 °C. Serum was collected and snap frozen in liquid nitrogen prior to being stored long-term at −80 °C. Serum was shipped on dry ice to IDEXX BioAnalytics (North Grafton, MA, USA) for analysis in their comprehensive chemistry panel (#6006).

### 2.20. Statistical Analysis

Animals were randomly assigned to either vehicle or treatment groups. All experimenters were blinded to the treatment group at the time of data acquisition, and were only unblinded after statistical analysis. All statistical analysis was done using GraphPad Prism 9 (GraphPad Software, LLC, San Diego, CA, USA). When comparisons were made between two normally distributed groups, a two-tailed, unpaired student’s t-test was performed. When comparisons were made between three or more groups, a one-way ANOVA with Tukey’s multiple comparisons test was performed. All results where *p* < 0.05 were considered statistically significant.

## 3. Results

### 3.1. Synthesis and Characterization of D-DPTIP

The synthesis of D-DPTIP was achieved using the highly efficient copper (I) catalyzed alkyne-azide click (CuAAC) reaction in three synthetic steps (Figure 1). CuAAC click has emerged as a highly versatile and robust tool for the construction of macromolecules and conjugation of bioactive ligands [60,61,62,63]. In the first step of the conjugate synthesis, the phenolic–OH group of DPTIP (compound **1**) was modified to attach an azide-terminating clickable linker to afford DPTIP-azide (compound **2**). This was achieved by reacting DPTIP with azido-PEG4-acid in the presence of EDC/DMAP at room temperature. The linker was attached onto DPTIP via an enzyme cleavable ester linkage to allow intracellular drug release. The structure of compound **2** was characterized by 1H NMR (Figure 2A and Appendix A) and ESI-MS (Appendix A).

In the second step, the surface hydroxyl groups on generation-4 hydroxyl polyamidoamine dendrimer (PAMAM-G4-OH; compound **3**) were partially modified with 5-hexynoic acid using standard esterification conditions (EDC/DMAP) to obtain D-Hexyne (compound **4**). The integration comparison of ester methylene protons at δ 4.01 ppm to dendrimer internal amide protons at δ 8.25–7.65 in 1H NMR suggested the attachment of an average ~7 alkyne arms (Figure 2 and Appendix A). The D-Hexyne was highly pure (>99%) as analyzed by the HPLC (Appendix A). In the third and final step of conjugate synthesis, D-Hexyne was reacted with DPTIP-azide using classical click conditions in the presence of catalytic amounts of copper-sulfate pentahydrate and sodium ascorbate to afford D-DPTIP conjugate (compound **5**). The use of the CuAAC reaction provided complete control on the drug loading that was evident through the synthesis of multiple 5 g scale batches of D-DPTIP. The 1H NMR spectrum clearly showed the presence of DPTIP protons along with the dendrimer protons (Figure 2A and Appendix A). The DPTIP loading was confirmed by the proton integration method by comparing the DPTIP protons in the aromatic region of the spectrum and DPTIP methoxy protons at δ 3.84 ppm to dendrimer internal amide H. The purity of the resulting D-DPTIP conjugate was >99% (Figure 2B and Appendix A). The hydrodynamic radius of D-DPTIP was ~6 nm with a near neutral zeta potential as analyzed by dynamic light scattering (Appendix A). The physicochemical properties of D-DPTIP conjugate are presented in Figure 2C.

The number of alkyne arms on D-Hexyne and the attachment of corresponding DPTIP molecules on the dendrimer surface were chosen to be kept at ~7 to maintain the overall water solubility of the conjugate due to the hydrophobic nature of DPTIP. Beyond this number, we started to observe aqueous insolubility of the dendrimer conjugate. The attachment of ~7 molecules of DPTIP on dendrimer surface corresponded to ~13% loading *w*/*w*. We have previously shown that a drug loading of <20% *w*/*w* on the dendrimer surface does not alter the pharmacokinetics of the dendrimer and maintains its brain targeting capabilities [44,48,54,55,64,65].

### 3.2. In Vitro Release of DPTIP from D-DPTIP

We further evaluated the drug release profile of the D-DPTIP conjugate to estimate the release of free DPTIP under physiological and endosomal conditions. To mimic the physiological conditions, the study was carried out at 37 °C with constant shaking. A phosphate buffer saline (PBS; pH 7.4) was used to mimic the plasma conditions. Since the intracellular uptake of PAMAM-OH dendrimers occurs via fluid phase endocytosis [66], we used the citrate buffer (pH 5.5), along with esterases, to mimic the endosomal conditions. At pH 7.4, we observed <20% release of DPTIP from the conjugates over a period of 3 weeks, suggesting that the D-DPTIP conjugate was stable in extracellular conditions. However, at intracellular conditions, more than 90% of DPTIP was released from the dendrimer conjugate over a period of 2 weeks, suggesting a sustained release profile (Figure 3A). At early time-points, within 24 h, negligible DPTIP release was observed at pH 7.4, and ~10% DPTIP was released in 24 h at intracellular conditions (Figure 3B).

### 3.3. Oral D-DPTIP Inhibits EV Release into Plasma Following Acute Brain Injury in Mice

We initially evaluated brain levels of DPTIP released from D-DPTIP in mice treated with 30 and 100 mg/kg (DPTIP equivalent) doses, and showed that 100 mg/kg afforded concentrations greater than the IC_50_ (32 ± 9.2 pmol/g), while 30 mg/kg gave low, variable levels (3.02 ± 1.5 pmol/g) (Appendix A). This dose was selected for all subsequent efficacy studies.

We next evaluated the efficacy of D-DPTIP in a murine acute brain injury model where intra-striatal IL-1β injection induces EV release into plasma. We utilized GFAP-GFP mice to monitor EV release from cells in the brain, specifically astrocytes, in addition to total plasma EV content, as previously described [27,29]. Mice were pretreated with a single oral 100 mg/kg D-DPTIP equivalent dose 48 h, 72 h, or 96 h prior to the striatal injection of IL-1β, or a single oral 100 mg/kg DPTIP dose or vehicle at 48 h prior to IL-1β injection (Figure 4A). We observed a significant increase in both total EVs (Figure 4B; *p* = 0.0066) and GFP + EVs (Figure 4C; *p* = 1.48 × 10^−5^), following IL-1β injection. D-DPTIP pretreatment led to a time-dependent inhibition of EV release. Total EVs decreased at both 48 h (*p* = 0.0091) and 72 h D-DPTIP pre-treatment times (*p* = 0.044), while pretreatment at 96 h did not reach statistical significance. A similar effect was observed with the GFP + EV release where 48 h (*p* = 0.00053) and 72 h (*p* = 0.0031) D-DPTIP pretreatment significantly reduced GFP + EVs, while pretreatment at 96 h did not reach statistical significance. In contrast, treatment with free DPTIP at 48 h prior to IL-1β injection did not result in significant inhibition of total or GFP + EVs. Based on these data, we selected a three times per week oral dosing regimen for D-DPTIP in the chronic AAV propagation model efficacy study.

### 3.4. Oral D-DPTIP Is Localized to Areas of Active Neuroinflammation

To directly measure the effect of D-DPTIP on EV propagation of pathological tau in the brain, our lab developed a rapid tau propagation model (AAV-hTau). Building upon prior propagation models [16,59,67], we utilized the hTau vector CBA-hTau24(P301L)(S320F)-WPRE (kindly provided by the Chakrabarty lab at the University of Florida) packaged into AAV1-serotype viral particles. 5 × 10^9^ viral particles, which contained human P301L/S320F double mutant tau, were injected unilaterally into the left hippocampus (AP-2.3, ML-2.1, DV-2.2) of 10-month-old C57BL6/J mice (Figure 5A). Six weeks following injection, mice were sacrificed, and their brains were stained for Thr181 phosphorylated tau and neurons (NeuN) (Figure 5B,C). We observed robust ipsilateral Thr181 tau staining in the neurons of the granule cell layer of the dentate gyrus (DG) and the pyramidal cell layer in the CA3 region of the hippocampus. Contralaterally, neurons in the hilus region of the dentate gyrus had significant Thr181 tau staining, with very little neuronal staining elsewhere. This is likely due to the direct connections between the hemispheres in mouse brains where the granule cell layer of the dentate gyrus connects contralaterally with cells in the hilus region (Allen Mouse Brain Connectivity Atlas, http://connectivity.brain-map.org/projection/experiment/112745787, accessed on 30 December 2020 [68,69]).

We next examined the biodistribution of orally administered fluorescently labelled dendrimer in the brain of AAV-hTau mice using dendrimers conjugated with Cy5 (D-Cy5). Previous studies using PAMAM-OH dendrimers have observed selective uptake by activated microglia in the brains of diseased or injured animals, while healthy animals had minimal uptake [48,49,50]. AAV-hTau mice were dosed orally with 55 mg/kg D-Cy5 6 weeks after AAV injection, and the brains were imaged 24 h post dose. In the frontal cortex, no significant D-Cy5 signal was observed both with minimal Iba1 staining and no Thr181 tau staining (Figure 6A–D). In contrast, robust D-Cy5 signal was observed at the site of AAV-hTau injection in the hippocampus with highly activated microglia and significant Thr181 tau staining in dentate gyrus neurons (Figure 6E–H). At a higher magnification, activated microglial cells were observed with robust D-Cy5 co-localization, indicating microglial uptake of the D-Cy5 (Figure 6I–K, white arrows).

### 3.5. Oral D-DPTIP Inhibits nSMase2 Activity in the Hippocampus and Is Well Tolerated

To further examine the localization and target engagement of D-DPTIP to nSMase2 in areas of neuroinflammation, we conducted nSMase2 enzymatic activity assessments in AAV-hTau mice treated for 6 weeks with D-DPTIP or empty dendrimer control (Figure 7A). nSMase2 activity assessments were completed 48 h following the final dose. Since sex differences have been reported in AD [70,71,72], we analyzed our data by sex. Male mice treated with D-DPTIP had significant and robust reduction of hippocampal nSMase2 activity (dark red vs. dark blue; *p* = 4.5 × 10^−7^) (Figure 7B). Similarly, female mice treated with D-DPTIP had significantly reduced nSMase2 activity (light red vs. light blue; *p* = 0.0013). In line with the selective localization of D-Cy5 to areas of injury and neuroinflammation, neither the male nor female mice treated with D-DPTIP had significantly reduced nSMase2 activity in the frontal cortex (Figure 7C).

### 3.6. D-DPTIP Significantly Inhibits Tau Propagation

The efficacy of D-DPTIP in reducing the propagation of Thr181 phosphorylated tau was next evaluated in the AAV-hTau mouse model. Ten-week old male and female mice were injected into the hippocampus with the AAV-hTau vector and randomly sorted into either empty dendrimer vehicle or 100 mg/kg D-DPTIP groups (n = 10/group/sex). Following six weeks of 3×/week oral dosing, animals were sacrificed, and their brains stained for Thr181 phosphorylated tau and neurons (NeuN). The mean fluorescence intensity (MFI) of the Thr181 tau staining in the contralateral DG hilus was quantified, and the levels normalized to the ipsilateral staining to account for variations in the AAV injection volume. MFI was also used to account for any tau accumulation in the axons and dendrites as not all the contralateral signal was contained solely in the cell bodies (Figure 5i). When we compared the average MFI of male mice treated with vehicle (0.114 ± 0.0892 MFI) with male mice treated with D-DPTIP (0.0475 ± 0.0180 MFI), we observed a 58% reduction in treated animals (Figure 8A–G; *p* = 0.034). Similarly, comparing female mice treated with vehicle (0.198 ± 0.0372 MFI) to D-DPTIP treated female mice (0.159 ± 0.0320 MFI) revealed a 20% reduction in treated mice (Figure 8I–O; *p* = 0.018). There were also significantly fewer Thr181 tau positive neurons in the contralateral DG hilus from mice treated with D-DPTIP in both the male (Figure 8A–F, H; *p* = 8.5 × 10^−5^) and female groups (Figure 8I–N, P; *p* = 1.0 × 10^−5^).

Importantly, chronic D-DPTIP dosing was well tolerated. Animals treated with D-DPTIP 3×/week for 6 weeks did not exhibit any overt toxicities or significant alterations in their body weight throughout the course of treatment (Appendix A). Additionally, clinical chemistry parameters measured at the end of the experiment did not differ from the normal ranges observed in our facility and those reported by Jackson Laboratories [73,74,75], Charles River [76], or Taconic [77] (Appendix A). There was a mild but significant elevation in blood urea nitrogen (BUN) levels in both the male and female D-DPTIP treated mice; however, the extent of the elevation was within the reference ranges from both healthy animals in our facility and those reported by others [73,74,75,76,77]. Therefore, we conclude that chronic D-DPTIP dosing was well tolerated in the mice throughout this experiment.

## 4. Discussion

The present studies demonstrate the utility of dendrimer-based delivery to improve the oral pharmacokinetic and brain penetration of the potent nSMase2 inhibitor DPTIP. We show that oral administration of D-DPTIP inhibits nSMase2 activity in the brain and blocks the increase in plasma EVs caused by acute brain injury up to 72 h post administration. We present a new tau propagation model in mice where human P301L/S320F double mutant tau plasmid (AAV-hTau) is unilaterally injected into the left hippocampus and taken up by the granule cell layer of the dentate gyrus, which, after several weeks, leads to an accumulation of hyperphosphorylated tau in the hilus region of the right hippocampus following known connectivity pathways. Employing this model, we show that chronic oral administration of D-DPTIP reduces the propagation of hyperphosphorylated tau across hemispheres. These data collectively demonstrate the preclinical efficacy of a dendrimer-conjugated nSMase2 inhibitor to block the propagation of toxic hyperphosphorylated tau species in mice, representing an exciting new therapeutic strategy for AD.

Tau propagation models facilitate the study of EV-mediated tau spread between interconnected regions in the mouse brain. Other research groups have demonstrated the feasibility of injecting an AAV vector expressing human mutant tau and monitoring its propagation to other interconnected regions of the brain [16,59,67]. For example, the Ikezu laboratory demonstrated that inhibiting EV biogenesis using GW4869, a tool nSMase2 inhibitor, reduced tau propagation from the entorhinal cortex, their site of injection, to the hippocampus, an interconnected region [16]. Our findings build upon this seminal work by demonstrating similar results with a different propagation model and a structurally distinct nSMase2 inhibitor that has significantly improved potency and PK properties. The combination of previous work and the present study strongly support targeting nSMase2 as a strategy for slowing tau propagation and thus the progression of AD.

DPTIP is the most potent nSMase2 inhibitor identified to date [29]. However, it suffers from poor pharmacokinetic properties (oral bioavailability < 5% and rapid clearance with a t_1/2_ < 0.5 h) [30,31]. Unfortunately, significant medicinal chemistry efforts around DPTIP did not result in any inhibitors with improved PK properties [31]. Conjugation of small molecule drugs to dendrimer nanoparticles have been posited as a mechanism to improve PK properties. In fact, hydroxyl PAMAM dendrimer conjugates discovered by our team are currently being tested in phase I (NCT05395624 and NCT03500627) and phase II clinical trials (NCT04458298), with positive Phase 2a results in severe COVID-19 patients in several measures of efficacy including survival, attenuation of neurological injury, and hyper-inflammation [53]. Several preclinical studies have demonstrated that dendrimer conjugation of small molecules can prolong their half-life in target tissues [44,45,46,47] and deliver drugs in a targeted manner to areas of neuroinflammation where they are engulfed by activated microglia [44,48,49,50]. Our data support the utility of dendrimer conjugation to improve the pharmacokinetics of small molecule inhibitors. First, we show that a single oral dose of D-DPTIP inhibits EV release in vivo up to 72 h post administration, which is significantly improved compared to DPTIP, which has a t_1/2_ of < 0.5 h [30,31]. Secondly, when equimolar doses are directly compared, oral D-DPTIP is significantly more effective than the unconjugated compound at inhibiting EV release (Figure 4B,C). Finally, oral D-DPTIP can deliver DPTIP brain levels above the IC_50_ in the brain for >24 h (Appendix A).

In addition to examining the efficacy of nSMase2 inhibition on tau spread in vivo, the AAV-hTau propagation model allowed us to investigate the biodistribution of D-DPTIP. We showed that in the AAV-hTau model, D-DPTIP specifically targeted areas of active neuroinflammation (hippocampus), while sparing unaffected bran regions (frontal cortex). This was supported by two lines of evidence. First, while significant D-Cy5 was observed in the hippocampus in the hTau-AAV mice, along with microglial activation and tau expression, very little D-Cy5 was found in the frontal cortex where there was no tau expression and minimal microglial activation. Second, while oral D-DPTIP significantly inhibited nSMase2 activity in the hippocampus of both male and female mice, it had no effect on nSMase2 activity in the frontal cortex. This latter observation is attributed to the lack of microglial activation in the cortex. Activity of nSMase2 in the presence of D-DPTIP were similar in the hippocampus and cortex; however, in the cortex, and distant from the site of insult, there was no increase in nSMase2 activity. This indicates that in order for there to be inhibitory effects, there needs to be an increase in nSMase2 activity driven by inflammation as a result of pathology, thus leading to targeted D-DPTIP localization. This is further substantiated by nSMase2 activity levels increasing in response to an inflammatory insult in our acute brain injury model (Figure 4B,C), as well as nSMase2 inhibition leading to reduced inflammatory markers of microglia [58,78], thus demonstrating a link between inflammation and nSMase2 activity.

Interestingly, there were observable sex differences in the D-DPTIP efficacy, where male mice had a greater reduction in tau propagation (58% vs. 20% decrease in contralateral MFI) and a reduced percentage of tau + neurons (41% vs. 32%) compared to females. These sex differences are important to consider since sex disparities in patients with AD have been well documented [70,71,72]. AD is more prevalent in women, and women typically have higher pathological severities than men [72], indicating that the disease burden may be higher in woman and more difficult to treat. Increased prevalence in woman may also be due to women living longer than men on average, since age is the greatest risk factor for AD [79]. Importantly, while serum ceramides were found to correlate with increased AD risk in a cohort of female patients [80], when comparing the risks of men and women, the Baltimore Longitudinal Study of Aging found that increased plasma ceramide levels were associated with an increased risk of AD in male patients but not in female patients [81]. That ceramides play a stronger role in disease burden in males may indicate why the male mice in our study responded more than female mice; however, this requires further study due to the complexities involved in understanding sex as a biological variable in AD.

## 5. Conclusions

We demonstrate that dendrimer conjugation significantly enhances the pharmacokinetics of DPTIP, improving its oral efficacy, brain penetration, and t_1/2_, and extending its time of efficacy up to 72 h post dose. Additionally, we show that oral D-DPTIP can selectively deliver DPTIP to areas with neuroinflammation in the brain, inhibit nSMase2 activity, and slow tau propagation in an hTau-AAV injection model. Given the poor potency, solubility, and pharmacokinetic profiles of known nSMase2 inhibitors [8,23,24,25,26], D-DPTIP represents a significant improvement in the field with translational potential for the treatment of AD.

## Figures and Tables

**Figure 1 pharmaceutics-14-02066-f001:**
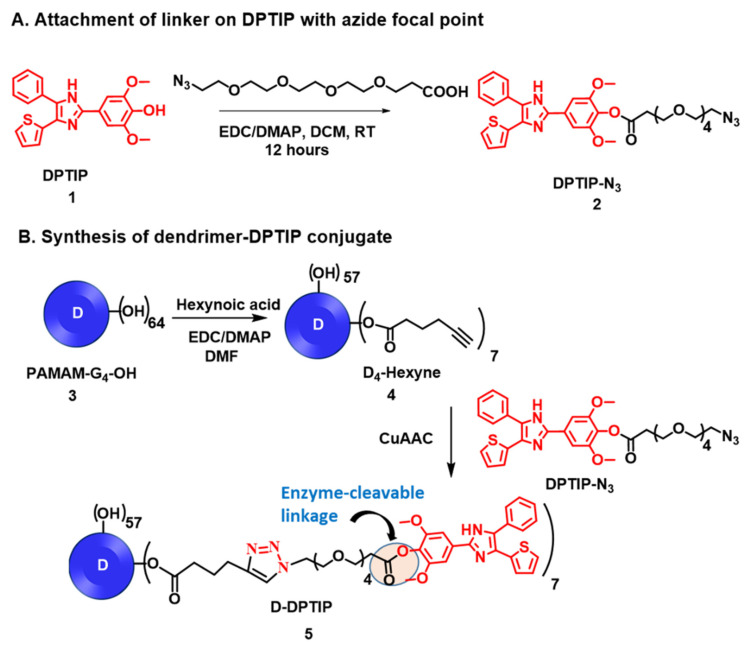
Schematic representation for the synthesis of D-DPTIP conjugate. (**A**) The modification of 2,6-Dimethoxy-4-(5-Phenyl-4-Thiophen-2-yl-1H-Imidazol-2-yl)-Phenol (DPTIP) to obtain a clickable DPTIP-azide; (**B**) synthesis of clickable dendrimer and D-DPTIP conjugate via CuAAC reaction.

**Figure 2 pharmaceutics-14-02066-f002:**
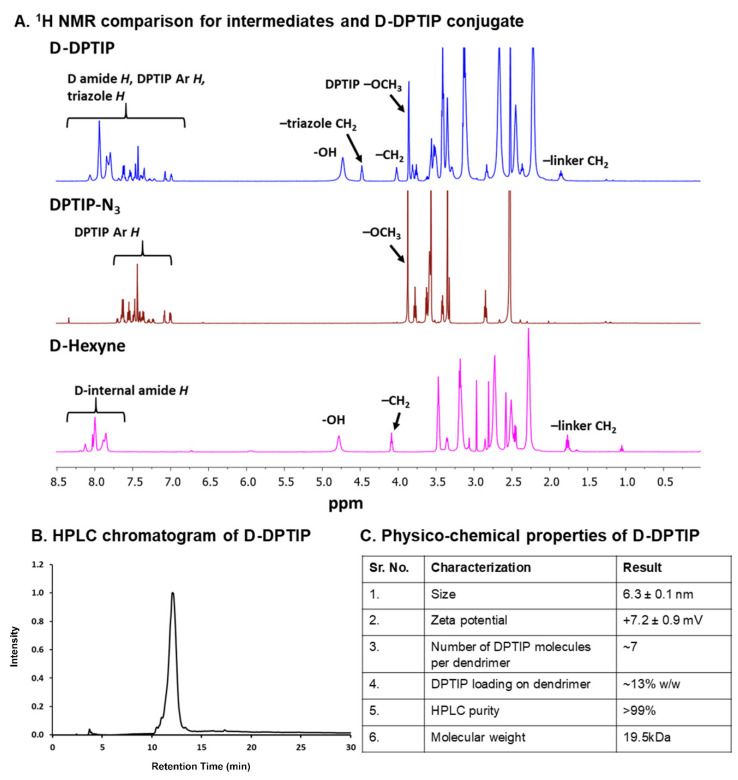
Physico-chemical characterization of intermediates and the D-DPTIP conjugate. (**A**) 1H NMR spectra showing the proton peaks of DPTIP-azide, and D-hexyne compared to the D-DPTIP conjugate. (**B**) HPLC chromatogram of D-DPTIP, retention time: 12.05 min, purity: 99.1%. (**C**) Table representing the physicochemical properties of the D-DPTIP conjugate.

**Figure 3 pharmaceutics-14-02066-f003:**
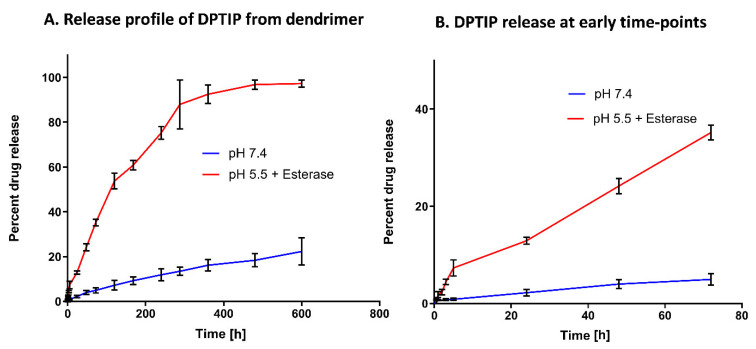
In vitro drug release profile of the D-DPTIP conjugate. (**A**) DPTIP shows a sustained release behavior at intracellular pH with >90% release in 2 weeks and only ~20% release at pH 7.4 in ~21 days. (**B**) The drug release profile at the early time points showing ~10% release in 24 h at intracellular conditions with a negligible DPTIP release at pH 7.4.

**Figure 4 pharmaceutics-14-02066-f004:**
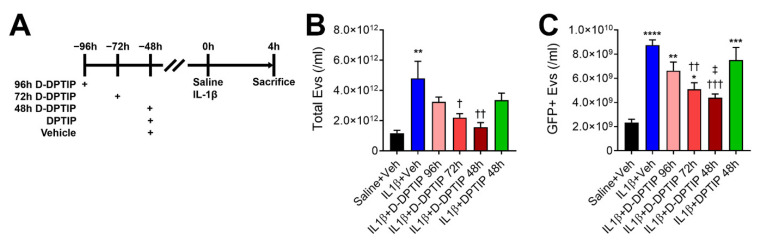
Oral D-DPTIP inhibits EV release into plasma in an acute brain injury model. (**A**) Experimental schematic. Mice were given a single oral dose of 100 mg/kg DPTIP, D-DPTIP, or vehicle at 48–96 h prior to intra-striatal IL-1β injection. Four hours later, mice were sacrificed and both total EV and GFP + EV levels in the plasma were assessed. (**B**) Quantification of total plasma EVs from intra-striatal saline + vehicle (black), intra-striatal IL-1β + vehicle (blue), intra-striatal IL-1β + 96 h D-DPTIP pretreatment (pink), intra-striatal IL-1β + 72 h D-DPTIP pretreatment (red), intra-striatal IL-1β + 48 h D-DPTIP pretreatment (dark red), and intra-striatal IL-1β + 48 h DPTIP pretreatment (green). (**C**) Concentration of GFP + EVs released into the plasma from astrocytes in the brain from the same animals shown in B. Bars represent mean ± SEM. * *p* < 0.05, ** *p* < 0.01, *** *p* < 0.001, and **** *p* < 0.0001 significant from saline + veh. † *p* < 0.05, †† *p* < 0.01, and ††† *p* < 0.001 significant from IL-1β + veh. ‡ *p* < 0.05 significant from IL-1β + DPTIP 48 h.

**Figure 5 pharmaceutics-14-02066-f005:**
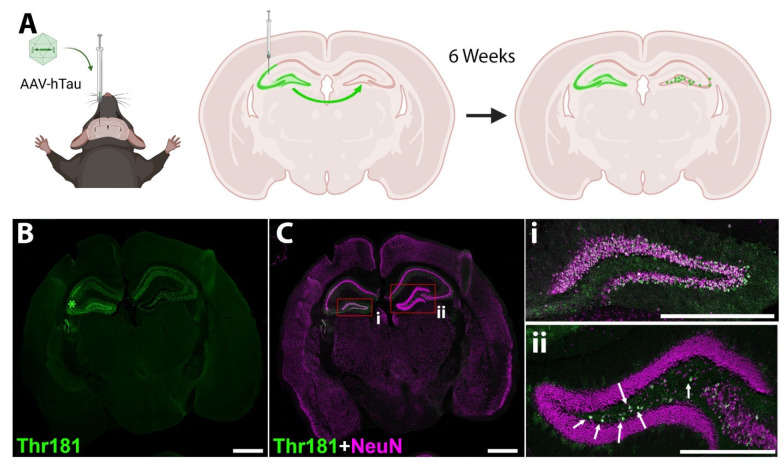
Murine model of mutant hTau propagation. (**A**) Schematic of the model. AAV vector expressing human P301L/S320F double mutant tau is injected into the left dorsal hippocampus near the CA3 region. The vector is taken up by hippocampal neurons, and after 6 weeks, phosphorylated tau expression is observed both in neurons of injection site (including hilus region of dentate gyrus) and neurons of the contralateral hilus region of the dentate gyrus (DG). Image was created with BioRender.com. (**B**) Representative image of Thr181 phosphorylated tau 6 weeks following AAV-hTau injection. Staining is observed in the left dorsal hippocampus near the CA3 region where injected (*). After six weeks, Thr181 phosphorylated tau staining is also observed in neurons of the contralateral hilus region of the dentate gyrus. (**C**) Merged image showing Thr181 tau (green) and neuronal NeuN (magenta) staining. Scale bar 1000 µm. (**i**) Inset from (**C**) showing the ipsilateral DG; (**ii**) inset from (**C**) showing the contralateral DG. Arrows indicate neurons that have taken up tau in the hilus region. Scale bar, 500 µm.

**Figure 6 pharmaceutics-14-02066-f006:**
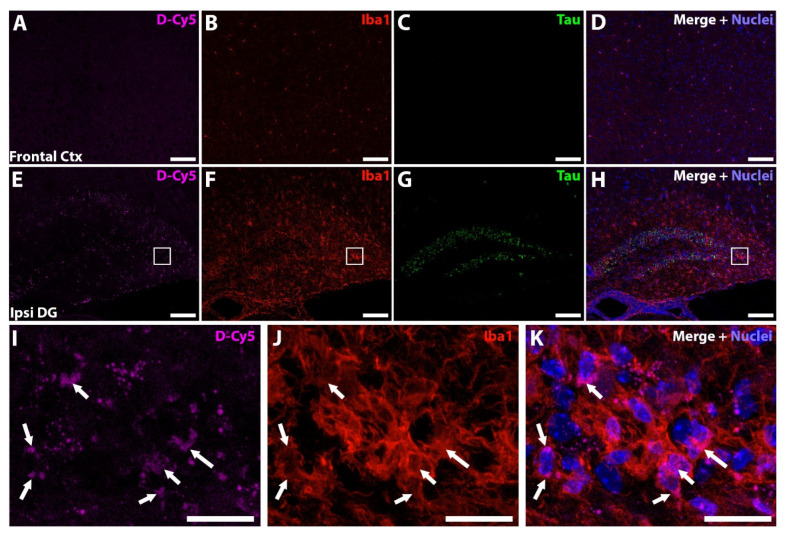
Dendrimer-Cy5 (D-Cy5) is targeted to areas of neuroinflammation. (**A**–**D**) Representative frontal cortex images showing no D-Cy5 signal (**A**, magenta), with low microglial activation (**B**, red), and no Thr181 tau expression (**C**, green). Merged image with nuclei stained blue (**D**). (**E**–**H**) Representative hippocampal images showing the localization of D-Cy5 (**E**) to areas of microglial activation (**F**) of the dentate gyrus with Thr181 tau expression (**G**). Merged image with nuclei stained blue (**H**). (**A**–**H**) Scale bar 100 µm. (**I**–**K**) Higher magnification of the white boxed region in (**E**–**G**). D-Cy5 signal (**I**) localized to microglial cells (**J**, white arrows). Merged image with nuclei stained blue shown in (**K**). (**I**–**K**) Scale bar 20 µm.

**Figure 7 pharmaceutics-14-02066-f007:**
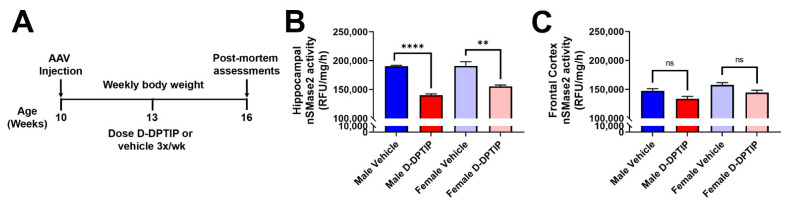
Orally administered D-DPTIP inhibits nSMase2 activity in the hippocampus of mutant hTau AAV injected mice. (**A**) Schematic highlighting the timing of dosing and data collection. (**B**) nSMase2 activity in the hippocampus of AAV-hTau mice 48 h following the terminal dose. Both male and female mice treated with D-DPTIP exhibited significant inhibition of nSMase2 activity. ** *p* < 0.01; **** *p* < 0.0001. (**C**) nSMase2 activity in the frontal cortex of AAV-hTau mice 48 h following the terminal dose. D-DPTIP treatment had no effect on nSMase2 activity in male and female mice. Bars represent mean ± SEM. Not significant (ns), *p* > 0.05.

**Figure 8 pharmaceutics-14-02066-f008:**
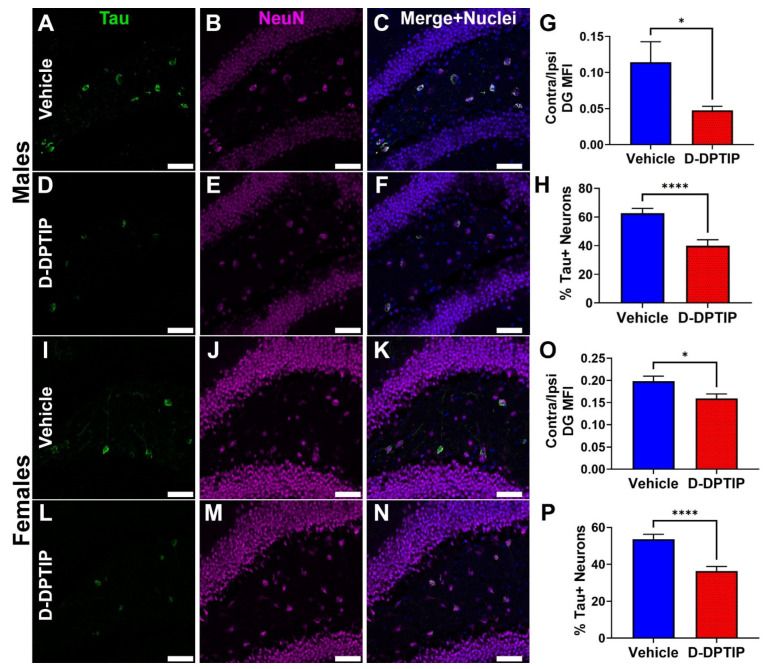
D-DPTIP inhibits the propagation of mutant hTau. (**A**–**F**) Representative images from male mice treated with empty dendrimer vehicle (**A**–**C**) or D-DPTIP (**D**–**F**) showing Thr181 tau staining in green (**A**,**D**), NeuN staining in magenta (**B**,**E**), and merged with nuclei stained blue (**C**,**F**). (**G**) Quantification of the mean fluorescence intensity of the ratio between contralateral and ipsilateral dentate gyrus Thr181 staining from male mice. (**H**) Quantification of the percentage of tau positive neurons in the contralateral hilus region of the DG from male mice. (**I**–**N**) Representative images from female mice treated with empty dendrimer vehicle (**I**–**K**) or D-DPTIP (**L**–**N**) showing Thr181 tau staining in green (**I**,**L**), NeuN staining in magenta (**J**,**M**), and merged with nuclei stained blue (**K**,**N**). (**O**) Quantification of the mean fluorescence intensity of the ratio between contralateral and ipsilateral dentate gyrus Thr181 staining from female mice. (**P**) Quantification of the percentage of tau positive neurons in the contralateral hilus region of the DG from female mice. Scale bar, 50 µm. Bars represent mean ± SEM. * *p* < 0.05. **** *p* < 0.0001.

## Data Availability

The data presented in this study are available within the manuscript and Appendix A.

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
