# Peer review of "Dendrimer-Conjugated nSMase2 Inhibitor Reduces Tau Propagation in Mice"

_pharmaceutics, 2022, doi:10.3390/pharmaceutics14102066_

Round 1

Reviewer 1 Report

The work is done with high quality and detail. Before the paper would be published I recommend a deeper discussion on DPTIP release when D-DPTIP conjugate was administrated in vivo. The role of dendrimer should be described in more details in Discussion and Conclusions sections. 

Author Response

The work is done with high quality and detail. Before the paper would be published I recommend a deeper discussion on DPTIP release when D-DPTIP conjugate was administrated in vivo.

  • This is an excellent suggestion. We have now included additional data on the levels of released DPTIP in the brain following two doses of D-DPTIP, as well as discussion of the same (see Supplemental Figure S9 and lines 459-463).

The role of dendrimer should be described in more details in Discussion and Conclusions sections. 

  • Agree. We have elaborated on the roles that dendrimer conjugation play in improving the pharmacokinetics of DPTIP in discussion (lines 646-654) and conclusions (lines 694-696).

Reviewer 2 Report

In figure 2B, define the X- and Y-axis titles.

In figure 7C, define the significance level by asterisks.

Author Response

In figure 2B, define the X- and Y-axis titles.

  • We thank the reviewer for catching this error and have added axis titles to figure 2B.

In figure 7C, define the significance level by asterisks.

  • We thank the reviewer for pointing out the inconsistencies with the figure legend. We have added "ns" to figure 7C to indicate that there were no significant differences between the groups. Furthermore, to increase clarity we moved the asterisks definitions to the 7B description and added a definition of ns to 7C.

Reviewer 3 Report

The study “Dendrimer Conjugated nSMase2 Inhibitor Reduces Tau Propa-2 gation in Mice” is well planned and executed. The study shows the improved effects of dendrimer conjugated drug over the normal. The experimental data well support the hypothesis. Here are a few comments that need to be addressed:

1.      How was the dose of the D-DPTIP finalized for oral administration?

2.      Mention the animal ethical committee approval in the M&M section.

3.      The empty dendrimer was given in drinking water in the vehicle group. So how the dose equivalency was ensured?

4.      Mention the anesthetics used to perform the brain surgery, if any.

5.       Check for typos, e.g. in line 406, correct 37oC etc.

Author Response

The study “Dendrimer Conjugated nSMase2 Inhibitor Reduces Tau Propagation in Mice” is well planned and executed. The study shows the improved effects of dendrimer conjugated drug over the normal. The experimental data well support the hypothesis. Here are a few comments that need to be addressed:

How was the dose of the D-DPTIP finalized for oral administration?

  • We chose 100mg/kg D-DPTIP as it was the dose that provided levels of brain DPTIP at its nSMase2 IC50; in contrast 30mg/kg D-DPTIP provided brain levels below the IC50.. We thank the reviewer for asking this important question,  and have added a supplemental figure showing these data (Figure S9) and a discussion has been added to lines (459-463).

Mention the animal ethical committee approval in the M&M section.

  • We have added the following statement to the methods (lines 188-191):
    • "All animal studies carried out in this manuscript were reviewed for ethical use of animals and approved by the Johns Hopkins University Animal Care and Use Committee (protocol # MO20M71; approved on 3/18/2020)."

The empty dendrimer was given in drinking water in the vehicle group. So how the dose equivalency was ensured?

  • We apologize for this lack of clarity.  Animals were orally gavaged with the empty dendrimer vehicle in the same manner as the Dendrimer-DPTIP.  They were not given ad libitum access to empty dendrimer vehicle water in place of regular water. We have updated the methods section to make this distinction clearer as follows (lines 206-207):
    • "DPTIP, D-DPTIP, and empty vehicle doses were given either as a single bolus (for EV release assays) or three times per week (for efficacy studies) via oral gavage."

Mention the anesthetics used to perform the brain surgery, if any.

  • We appreciate the reviewer catching this omission and have added this to (line 237):
    • "...mice were deeply anesthetized with 1-2% isoflurane..."

Check for typos, e.g. in line 406, correct 37oC etc.

  • We have carefully gone through the manuscript and fixed several identified typos and grammatical errors.
